# DocRobust: Enhancing Robustness of Multimodal LLMs in Low-Quality Document Image Scenarios

## Abstract

Document images are primary carriers of knowledge and information, yet their effective understanding is often hindered by degradations such as noise, blur, and low resolution. In this paper, we address the challenge of robust document understanding under such low-quality conditions by proposing the DocRobust-Module (DRM)—an efficient feature restoration module that, when integrated with a multimodal large language model, enables the recovery of lost visual and semantic information with minimal parameter modifications. Our method is supported by a novel two-stage training strategy that incrementally guides the model to restore critical information from both visual and semantic perspectives. To support the fine-tuning of MLLMs with DRM, we construct DocRobust-VQA, a large-scale visual question answering dataset containing extensive low-quality document images along with high-quality counterparts and QA annotations. With over 189K clear-blurry images pairs annotated by 417K QA pairs, DocRobust-VQA provides sufficient finetuning data for enhancing the robustness of MLLMs under real-world degradations. Extensive experiments demonstrate that our method consistently improves performance on low-quality document images, offering new insights and a scalable solution for robust document understanding.

## 1 Introduction

Documents are primary carriers of knowledge and information, and their practical parsing and comprehension are crucial for improving information processing efficiency and enabling digital workflows Xu et al. (2020b). Document understanding has demonstrated significant value in various fields, including information extraction, text recognition, and knowledge management.

In recent years, Multimodal Large Language Models (MLLMs) have advanced rapidly, and document understanding has emerged as a key application Liu et al. (2024); Luo et al. (2024); Hu et al. (2024); Ye et al. (2023). Capabilities such as text recognition and visual document question answering have gradually become major optimization objectives Liao et al. (2023); Blecher et al. (2023); Wang et al. (2024b). The continuous progress in these models provides robust technical support for document understanding, enabling cross-modal information fusion, and promoting a shift from traditional single-modal processing to comprehensive multimodal interpretation. However, in practical scenarios, document images often suffer from noise, blur, low resolution, and other degradations that lead to significant recognition and comprehension errors Das et al. (2019); Zhang et al. (2024a); Lin et al. (2020).

Although recent benchmarks such as R-bench Li et al. (2024a) and WildDoc Wang et al. (2025) have been introduced to evaluate the robustness of large models under low-quality visual conditions, there remains a significant gap in methods specifically designed for enhancing the robustness of MLLMs via low-quality image restoration. We attribute this gap to two main factors. On the one hand, while benchmarks for evaluation are becoming available, MLLMs typically require large-scale datasets for effective fine-tuning, and existing low-quality document image datasets are far from sufficient in scale or diversity. On the other hand, existing methods for low-quality document image restoration Zhang et al. (2024a); Souibgui & Kessentini (2020) typically focus on pixel-level recovery. Such dense supervision may lead to overfitting and, coupled with repetitive visual feature

extraction and dense pixel prediction, incurs substantial computational overhead—rendering joint optimization with document understanding tasks difficult.

To address these challenges, this paper focuses on enhancing the robustness of multimodal large language models in low-quality document image scenarios by tackling two core issues: (1) effectively improving the model's ability to recognize and understand low-quality images while minimizing changes to the overall model parameters, and (2) constructing a large-scale dataset that includes diverse low-quality document images to support robustness-oriented training.

To this end, our work comprises two main components:

**Model:** To mitigate performance degradation on low-quality images, we propose a feature restoration module, **DocRobust-Module (DRM)**. With minimal parameter adjustments, DRM effectively recovers corrupted visual and semantic information, thereby enhancing model robustness on degraded document images. To guide the module in learning effective restoration capabilities, we introduce a two-stage training strategy that encourages the model to recover lost information from both visual and semantic perspectives. Experimental results demonstrate that our proposed module consistently improves performance across the two training stages, including standalone training and joint optimization with the multimodal model.

**Data**: We construct a large-scale Visual Question Answering (VQA) dataset, **DocRobust-VQA**, which includes a broad range of low-quality document images collected from various real-world scenarios. The dataset consists of 189,771 images paired with 417,502 question-answer pairs, providing sufficient scale and diversity to support the training of multimodal large language models to gain robustness under degraded visual conditions.

In summary, the main contributions of this work are as follows:

- We propose an efficient feature restoration module, **DocRobust-Module (DRM)**, along with a two-stage training strategy, to restore and supplement lost information in low-quality document images, thereby enhancing the overall robustness of multimodal large language models.

- We introduce **DocRobust-VQA**, a large-scale dataset tailored for VQA on low-quality document images that provides rich diversity and sufficient data volume to facilitate both training and robustness evaluation of multimodal large language models under degraded visual conditions.

- We conduct a comprehensive comparative analysis of existing multimodal large language models on low-quality document images across various dataset and benchmark. Extensive experiments show that, after training on Docrobust-VQA, our proposed DRM not only enhances the robustness of MLLMs on the synthetic test sets, but also improves their performance on real-world degraded images and even provides a degree of robustness against adversarial examples, offering new insights and methodologies for robustness research in multimodal large language models.

## 2 RELATED WORKS

### 2.1 VISUAL DOCUMENT UNDERSTANDING

Visual document understanding (VDU), a key task in cross-modal learning, has evolved through three main stages. Early works Xu et al. (2020b;a); Huang et al. (2022); Li et al. (2021a;b); Gu et al. (2021); Appalaraju et al. (2021) focused on pretraining models that combine OCR-extracted text with layout features, achieving strong performance in structured document tasks. To address OCR-related limitations, later methods Kim et al. (2022); Davis et al. (2022); Tang et al. (2023); Lee et al. (2023) adopted end-to-end architectures that directly extract semantics via visual encoders. Recently, multimodal large language models (MLLMs) Luo et al. (2024); Liu et al. (2024; 2023); Wang et al. (2024a); Lu et al. (2024); Chen et al. (2024); Hu et al. (2024); Li et al. (2024c); Ye et al. (2023) pretrained on massive datasets have set a new paradigm, showing strong zero-shot and instruction-following capabilities. Although robustness under low-quality inputs has gained attention Li et al. (2024a), and datasets like WildDoc Wang et al. (2025) simulate real-world conditions,

there remains a lack of effective, MLLM-compatible restoration methods for degraded document images.

## 2.2 Methods for Degraded Document Images

Existing methods for handling degraded document images fall into two categories: data quality enhancement and model robustness improvement.

Data enhancement methods typically focus on image restoration, either targeting specific degradations Das et al. (2019); Lin et al. (2020); Wang et al. (2022); Yang et al. (2024); Zhang et al. (2022; 2023a;b) or using unified architectures Souibgui & Kessentini (2020); Souibgui et al. (2023); Yang et al. (2023) that still require task-specific training and inference. Recent work like DocRes Zhang et al. (2024a) supports multiple restorations via visual prompts, but such pixel-level methods lack semantic-level optimization and are inefficient for downstream tasks.

Robustness-oriented methods improve model tolerance via training strategies, including masked pretraining Lyu et al. (2022), contrastive learning Yang et al. (2022); Guan et al. (2023), and degradation simulation Wei et al. (2024). DoCo Li et al. (2024b) enhances visual encoding for dense text. However, these methods often rely on modifying pretraining objectives, limiting compatibility with existing MLLMs.

To overcome this, we propose the DocRobust Module (DRM) and a two-stage training strategy that restores feature-level quality without altering the backbone model, significantly boosting robustness for pretrained MLLMs under low-quality document conditions.

## 3 Dataset Construction

To enhance the robustness of multimodal large vision-language models on low-quality document images, we construct a DocRobust-VQA, composed of paired clear document images and their corresponding corrupted versions.

### 3.1 Clean Data Collection

The quality and quantity of clear images in the training set form the foundation of the dataset and are critical for enabling the multimodal model to learn robustness. In selecting clear document images, we considered the following factors: (1) Domain Diversity: The model should generalize across varied image types (e.g., office documents, receipts, charts, scene texts, and text line crops), therefore the training set incorporates diverse sources. (2) Scale Diversity: Perturbations affect images differently depending on dimensions and text sizes. To ensure robustness, the training set includes both large-format documents with small text and smaller crops or scene texts with larger fonts. (3) Clarity: While real data may contain degraded samples, paired training requires fully legible images to provide high-quality ground truth for restoration. Hence, our clean data ensure superior quality compared to corrupted inputs. Based on these considerations, we integrate and filter clear images with high-quality question-answer annotations from eight datasets, including ChartQA Masry et al. (2022), DT-VQA Zhang et al. (2024b), EST-VQA Wang et al. (2020), Single-page DocVQA Mathew et al. (2021), Multi-page DocVQA Tito et al. (2023), InfographicVQA Mathew et al. (2022), TextVQA Singh et al. (2019), and OCRBench_v2 Fu et al. (2024).

### 3.2 Corrupted Data Construction

Corrupted images are not only used to train the restoration model together with clear images but also serve as fine-tuning data for the multimodal model's SFT, using the question-answer annotations from the clear images. Moreover, when evaluating the multimodal model's low-quality image understanding ability, the benchmark is constructed from these corrupted images. Thus, the method for generating corrupted images is doubly important for enhancing and assessing the model's capabilities.

Based on the clear images collected in the previous section, we generate corrupted images. Specifically, we classify corruptions into five categories according to their visual effects: (1) Luminance, (2) Distortion, (3) Blurriness, (4) Noise and (5) Compression. For a given clear image, we randomly

select $k$ categories from these five, then randomly choose one specific corruption from each selected category, and finally apply the $k$ chosen corruptions sequentially to form the corrupted image. It is noteworthy that these five categories of corruption have a specific sequential order when applied to an image. We observed that some corruption effects can overlap or override others (e.g., Gaussian blur may obscure noise), so the order is fixed as listed above. Additionally, for each corruption, the strength is adjusted based on the image size (e.g., the size of the Gaussian kernel, the radius for flexible distortion, etc.).

## 4  METHOD

Based on the training data and benchmark constructed previously, we propose the **DocRobust-Module (DRM)**, a simple yet effective visual token restoration module. DRM restores corrupted tokens between the Visual Encoder and the Projector MLP, mapping tokens encoded from low-quality (LQ) images into a high-quality (HQ) space that is easier to parse, thereby enhancing the robustness of multimodal large language models on LQ images. In the following, we first describe how the DocRobust-Module is integrated with the multimodal large model, then detail its internal architecture, and finally introduce our two-stage training strategy.

### 4.1  OVERALL ARCHITECTURE

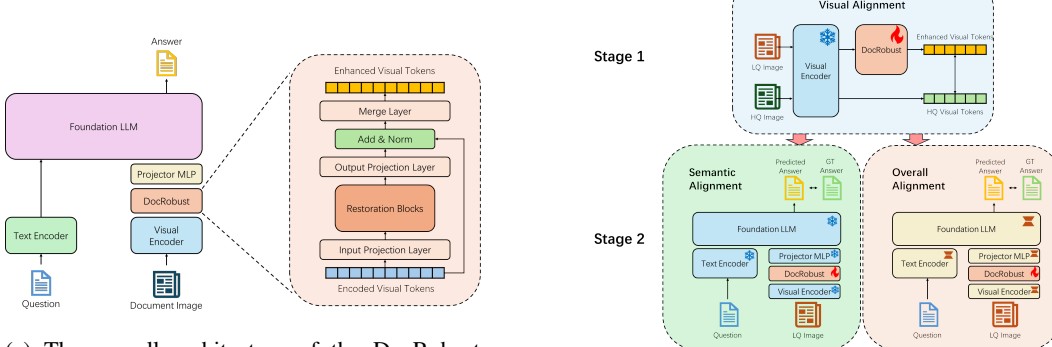

(a) The overall architecture of the DocRobust framework.

(b) The training pipeline of our two-stage strategy.

Figure 1: Overview of the DocRobust framework and its two-stage training strategy.

Current state-of-the-art multimodal large language models Wang et al. (2024a); Chen et al. (2024); Lu et al. (2024) typically consist of four components: a Foundation LLM, a Text Encoder, a Visual Encoder, and a Projector MLP. The Foundation LLM, usually built on an existing language model and trained in an autoregressive manner, is the primary source of the model's comprehension and reasoning. The Text Encoder maps input text into token sequences via an embedding layer and a tokenizer. The Visual Encoder converts input images into token sequences, typically implemented using a Vision Transformer (ViT) Dosovitskiy et al. (2021). The Projector MLP then maps these encoded visual tokens into a feature space aligned with the Foundation LLM, enabling effective multimodal reasoning between images and text.

However, when processing LQ images, the Visual Encoder or the Projector MLP cannot effectively recover the information lost due to image degradation, leading to misinterpretation by the multimodal model and reduced robustness. To address this, we insert a lightweight plug-and-play module—**DocRobust-Module (DRM)**—between the Visual Encoder and the Projector MLP to restore the encoded visual tokens by supplementing missing critical information.

As shown in Fig. 1a, given an input image $I \in \mathbb{R}^{H \times W \times C_{img}}$, the Visual Encoder produces encoded visual tokens $F_{vis} \in \mathbb{R}^{L \times D_{visual}}$. Due to operations such as Pixel Shuffle that may reduce the token sequence length, we have

$$L = \frac{H}{h_{patch}} \times \frac{W}{w_{patch}} \times S^2, \tag{1}$$

and the embedding dimension is

$$D_{visual} = \frac{D_{encoder}}{S^2},$$  (2)

where $w_{patch}$, $S$, and $D_{encoder}$ denote the patch size of the Visual Encoder, the downsampling ratio of the Pixel Unshuffle, and the encoder's embedding dimension, respectively. The encoded tokens $F_{vis}$ are then fed into DRM for restoration, yielding enhanced visual tokens $F_{enh} \in \mathbb{R}^{L \times D_{visual}}$. Finally, $F_{enh}$ is input to the Projector MLP for mapping and then passed into the Foundation LLM for inference.

## 4.2 DOCROBUST-MODULE

The DocRobust-Module (DRM) consists of five components: Input Projection Layer, Restoration Blocks, Output Projection Layer, Add & Norm Layer, and Merge Layer.

Both the Input Projection Layer and the Output Projection Layer are implemented as single linear layers with bias, while the Merge Layer comprises two linear layers. The Restoration Blocks are incorporated to effectively supplement missing information within the constrained feature dimensions. For simplicity, we adopt $N$ stacked Transformer blocks as the Restoration Blocks. Specifically, $F_{vis}$ is first processed by the Input Projection Layer, reducing its dimension to $D_{dr}$, then passed through the Restoration Blocks. The Output Projection Layer maps the resulting features back to a space consistent with $F_{vis}$, yielding supplementary features $F_{sup}$. Finally, the Add & Norm Layer and the Merge Layer integrate $F_{sup}$ into $F_{vis}$ to produce the enhanced visual tokens $F_{enh}$. This process is formulated as follows:

$$
\begin{aligned}
F_{sup} &= \text{Linear}_{\text{out}}(\text{RBs}(\text{Linear}_{\text{in}}(F_{vis}))), \\
F_{merge} &= \text{LayerNorm}(F_{sup} + F_{vis}), \\
F_{enh} &= \text{Linear}_1(\text{GELU}(\text{Linear}_0(F_{merge}))).
\end{aligned}
$$  (3)

Here, $\text{RBs}(\cdot)$, $\text{Linear}_{\text{in}}(\cdot)$, and $\text{Linear}_{\text{out}}(\cdot)$ denote the Restoration Blocks, the Input Projection Layer, and the Output Projection Layer, respectively.

## 4.3 TRAINING STRATEGY

To further enhance the overall robustness of the model on low-quality document images, we design a two-stage training strategy. The first stage performs **Visual Alignment**, then one of **Semantic Alignment** and **Overall Alignment** is applied in the second stage .

### 4.3.1 VISUAL ALIGNMENT

In the initial training phase, it is essential to endow DRM with effective visual restoration capabilities for LQ document images. Thus, we propose the Visual Alignment stage. By leveraging paired LQ and HQ images from our dataset, we guide DRM to restore LQ visual tokens to match as closely as possible their HQ counterparts. Unlike conventional pixel-level restoration methods, DRM directly restores visual tokens, reducing computational cost and mitigating overfitting risks associated with dense pixel-level supervision. Moreover, the enhanced visual tokens can be directly used in downstream inference without re-extraction. Specifically, as illustrated in Fig. 1b, given an LQ image $I_{LQ} \in \mathbb{R}^{H \times W \times C_{img}}$, and its corresponding HQ image $I_{HQ} \in \mathbb{R}^{H \times W \times C_{img}}$, the Visual Encoder (with fixed weights) extracts encoded visual tokens $F_{vis}^{LQ}$ and $F_{vis}^{HQ}$. The LQ tokens $F_{vis}^{LQ}$ are then processed by DRM to obtain $F_{enh}^{LQ}$. The objective during Visual Alignment is to minimize the discrepancy between $F_{enh}^{LQ}$ and $F_{vis}^{HQ}$:

$$
\begin{aligned}
F_{enh}^{LQ} &= \text{DRM}(F_{vis}^{LQ}), \\
L_{visual} &= \text{MSE}(F_{enh}^{LQ}, F_{vis}^{HQ}).
\end{aligned}
$$  (4)

### 4.3.2 SEMANTIC ALIGNMENT

After establishing visual restoration capabilities, DRM is further trained to recover semantic information critical for document understanding. In this stage, as shown in Fig. 1a, we integrate DRM into an existing multimodal large language model and perform Supervised Fine-Tuning (SFT) on our

low-quality document image VQA dataset while keeping the multimodal model parameters frozen. This process guides DRM to enhance its semantic restoration ability. The loss function for Semantic Alignment is given by:

$$L_{semantic} = \text{CrossEntropy}(Y_{semantic}, \hat{Y}), \tag{5}$$

where $Y_{semantic}$ and $\hat{Y}$ denote the model output and the corresponding ground truth, respectively.

### 4.3.3 OVERALL ALIGNMENT

In the Overall Alignment stage, we jointly fine-tune all modules of the multimodal large language model along with DRM to further improve robustness on LQ document images. SFT is conducted on the low-quality document image VQA dataset, and to reduce training costs, we apply LoRA fine-tuning Hu et al. (2022) to all parameters except those in DRM. The training objective during Overall Alignment is formulated as:

$$L_{overall} = \text{CrossEntropy}(Y_{overall}, \hat{Y}), \tag{6}$$

where $Y_{overall}$ represents the final output of the model in this stage.

The overall loss function for our two-stage training strategy is thus:

$$L_{all} = L_{visual} + L_{semantic} \text{ or } L_{all} = L_{visual} + L_{overall}. \tag{7}$$

## 5 EXPERIMENTS

### 5.1 IMPLEMENTATION DETAILS

For model architecture, we validate the effectiveness of DRM and the training strategy on the InternVL-2.5 series Chen et al. (2025). Specifically, we use the InternViT-300M Chen et al. (2024) visual encoder with an embedding dimension of $D_{encoder} = 1024$, an image block size of 448, and a patch size of 14. The Pixel Unshuffle downsampling ratio is set to $S = 0.5$, resulting in encoded visual tokens $F_{vis}$ with sequence length $L = 256$ and embedding dimension $D_{visual} = 4096$. In DRM, the number of Restoration Blocks (RBs) is set to $N = 6$, with an intermediate feature dimension of $D_{dr} = 512$, and the feedforward dimension of the Transformer blocks is 2048.

For model training, in the Visual Alignment stage, the model is trained for 5 epochs with a batch size of 256. We adopt an initial learning rate of 0.001, use a linear warmup over 0.5 epochs, and gradually decrease the learning rate according to a 1-cycle learning rate schedule. In the Semantic Alignment stage, the batch size is set to 128, with other hyperparameters following the standard SFT settings of InternVL-2.5. In the Overall Alignment stage, the LoRA rank is set to 128, and the remaining hyperparameters adhere to the standard LoRA SFT settings of InternVL-2.5. All model training and inference are performed on 4 NVIDIA L40S GPUs.

### 5.2 PERFORMANCE ON LOW-QUALITY SCENARIOS

To evaluate the robustness of multimodal large models on low-quality document images, we selected mainstream closed-source models (GPT4o Team (2024) and Gemini1.5-pro Team et al. (2024)) as well as open-source models (Qwen2.5-VL Bai et al. (2025) and InternVL-2.5 Chen et al. (2025)) as our primary comparison and analysis targets. For the closed-source models, scores on the standard datasets are taken from their public technical reports, whereas the scores on the corrupted data and the standard dataset scores for open-source models are computed using VLMEvalKit Duan et al. (2024).

### 5.2.1 RESULTS ON DOCROBUST-VQA

As shown in the Table 1, we compare the standard scores of the models on both the standard and corrupted datasets. Overall, all models exhibit a noticeable drop in scores on the corrupted data, which highlights the challenging nature of our proposed DocRobust-VQA.

Analyzing different subsets, we find that ChartQAMasry et al. (2022) suffers the most severe performance drop due to distortion-induced deformation of image lines, which impairs chart interpretation.

| Method | Tasks | Datasets | | | | |
|---|---|---|---|---|---|---|
| | | ChartQA | TextVQA | DocVQA | InfographicVQA | OCRBench |
| GPT4o | clean | 85.7 | 77.4 | 92.8 | 79.2 | 736 |
| | corrupted | 25.16 | 56.74 | 42.97 | 34.95 | 522 |
| Gemini1.5-pro | clean | 87.2 | 78.7 | 93.1 | 80.1 | 754 |
| | corrupted | 29.60 | 57.00 | 74.19 | 45.09 | 541 |
| Qwen2.5-VL-Max | clean | 88.48 | 81.46 | 95.74 | 81.84 | 805 |
| | corrupted | 58.40 | 62.96 | 85.93 | 59.05 | 597 |
| InternVL-2.5-4B | clean | 84.0 | 76.8 | 91.6 | 72.1 | 828 |
| | corrupted | 42.6 | 61.0 | 79.1 | 48.4 | 565 |
| InternVL-2.5-1B | clean | 76.24 | 72.01 | 84.75 | 55.75 | 788 |
| | corrupted | 34.16 | 56.95 | 72.11 | 37.44 | 563 |
| DocRobust-visual | clean | 75.60 | 71.53 | 84.49 | 56.33 | 783 |
| | corrupted | 44.80 | 56.55 | 72.55 | 37.34 | 566 |
| DocRobust-semantic | clean | 75.76 | 72.89 | 84.44 | 55.58 | 788 |
| | corrupted | 49.44 | 58.87 | 73.39 | 38.12 | 559 |
| DocRobust-overall | clean | 75.92 | 72.91 | 84.77 | 56.01 | 787 |
| | corrupted | 57.92 | 60.14 | 76.26 | 41.10 | 584 |

Table 1: Results on DocRobust-VQA. Docrobust-visual, Docrobust-semantic, and Docrobust-overall correspond to the InternVL-2.5-1B models integrated with DRM and trained through the Visual Alignment, Semantic Alignment, and Overall Alignment stages, respectively.

InfographicVQAMathew et al. (2022) also shows a notable decline, likely due to its chart-like content. In contrast, DocVQAMathew et al. (2021) and TextVQASingh et al. (2019) are less affected by distortion, with performance mainly degraded by blur and noise that obscure semantic content. These results highlight the differing sensitivities of visual and textual tasks, validating our dual alignment training strategy.

From the perspective of model performance, the latest open-source multimodal large models have reached or even surpassed the closed-source models on OCR-related standard datasets, thanks to their well-curated document understanding data and dynamic patch processing strategies for high-resolution image inputs. Notably, our method demonstrates a significant advantage in low-quality scenarios, which confirms the effectiveness of our proposed dataset and methods.

### 5.2.2 RESULTS ON REAL-WORLD DATASET

| Model | WildDoc | RealDoc-Clean | RealDoc-Corrupted |
|---|---|---|---|
| InternVL2.5-1B | 30.85 | 46.89 | 24.10 |
| Docrobust-overall | **36.69** | **52.91** | **32.09** |

Table 2: Average scores on WildDoc and RealDoc.

To further validate the robustness enhancement brought by our proposed DocRobust framework, we conducted experiments on real-world datasets using the InternVL2.5 model with and without the DocRobust module. Specifically, we selected two real datasets: the RealDoc dataset constructed by us, and the publicly available WildDoc dataset. The RealDoc dataset was built by filtering high-quality and low-quality image pairs from two real document image datasets, Inv3dReal Hertlein et al. (2023) and DocUNet Ma et al. (2018). For each image pair, we used Gemini 2.5-Pro to generate question-answer (QA) pairs grounded in the image content, resulting in a total of 2,141 annotated QA pairs. Meanwhile, WildDoc consists of over 12,000 low-quality document images captured in real-world scenarios, with image sources drawn from widely used document datasets such as DocVQA, ChartQA, and others. As shown in Table 2, the experimental results demonstrate that DocRobust significantly improves the robustness of MLLMs even on real-world datasets. This provides further evidence of the effectiveness of our DRM module, and shows that training on our proposed DocRobust-VQA dataset can indeed equip models with strong robustness capabilities.

### 5.3 PERFORMANCE ON ADVERSARIAL EXAMPLES

Remarkably, we observe that even in the absence of any adversarial training or explicit defense mechanisms, training on DocRobust-VQA alone enables the DRM module to significantly improve the model's resilience against adversarial attacks. As presented in Table 3, we evaluate adversarial robustness on four representative datasets—ChartQA, TextVQA, DocVQA, and InfographicVQA—by applying the MF-Attack Zhao et al. (2023) method to generate adversarial samples targeting the original InternVL2.5-1B model. Since the attack is a white-box attack specifically targeting InternVL2.5-1B, the model exhibits a substantial performance degradation under on these adversarial examples. Nonetheless, when equipped with the DocRobust module, the model demonstrates a notable improvement in adversarial robustness, despite having never seen adversarial examples during training. This suggests that the DRM module, through exposure to diverse degraded inputs in DocRobust-VQA, can implicitly enhance the model's robustness to perturbations beyond the training distribution.

| Method | Datasets | | | |
|---|---|---|---|---|
| | ChartQA | TextVQA | DocVQA | InfographicVQA |
| InternVL2.5-1B | 12.72 | 15.77 | 18.31 | 9.23 |
| Docrobust-overall | **17.00** | **24.96** | **29.22** | **14.33** |

Table 3: Results on the adversarial examples generated from ChartQA, TextVQA, DocVQA and InfographicVQA datasets.

| Method | Datasets | | | |
|---|---|---|---|---|
| | ChartQA | TextVQA | DocVQA | InfographicVQA |
| DiffBIR | 9.96 | 28.08 | 14.24 | 17.67 |
| DocRes | 28.76 | 58.11 | 70.85 | 37.09 |
| DocRobust-overall | **57.92** | **60.14** | **76.26** | **41.10** |

Table 4: Results on ChartQA, TextVQA, DocVQA and InfographicVQA datasets, applying different restoration methods.

### 5.4 COMPARISON TO PIXEL-LEVEL RESTORATION METHODS

To further validate the effectiveness and superiority of our proposed feature-level restoration method, we conducted comparative experiments between DocRobust and two representative pixel-level restoration methods on the ChartQA, TextVQA, DocVQA, and InfographicVQA datasets. Specifically, we compared against DiffBIR Lin et al. (2024), a general-purpose image restoration method, and DocRes Zhang et al. (2024a), a method tailored for document image enhancement. Additionally, we fine-tuned DocRes on our DocRobust-VQA dataset to ensure a fair comparison. For both pixel-level restoration methods, the evaluation pipeline is feeding the degraded images into the restoration module, and then passing the restored images to the original InternVL2.5 model for answering. The results, as shown in Table 4, demonstrate that even the fine-tuned DocRes performs significantly worse than DocRobust, with DiffBIR falling even further behind. These findings underscore the advantage of feature-level restoration over pixel-level restoration in the context of multimodal understanding under low-quality conditions, and highlight the efficacy of our proposed DocRobust.

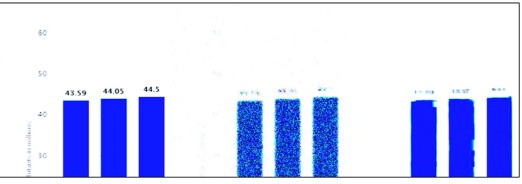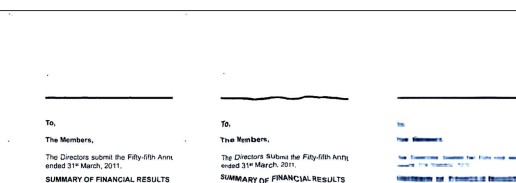

Figure 2: Visualization of reconstructed region from DRM after Visual Alignment training. Each group consists of three images, arranged from left to right as: HQ image, LQ image, and reconstructed region indicated by an additionally trained decoder.

### 5.5 EFFECTIVENESS OF TRAINING STRATEGY

We use InternVL2.5-1B as the base model to conduct ablation experiments on the two-stage training strategy proposed in this paper. As shown in Table 1, after the Visual Alignment stage, integrating DRM into the untrained base model yields a 10.6% improvement in the standard score on ChartQA—a dataset that emphasizes visual information for chart understanding—while other text-dominant datasets exhibit no significant change. This confirms that the Visual Alignment stage effectively guides DRM to learn visual information supplementation.

After the Semantic Alignment stage, further improvements are observed on ChartQA with an incremental gain of 15.2%, and enhancements are also recorded on other text-based VQA datasets. This indicates that Semantic Alignment enables DRM to acquire semantic information restoration capabilities.

After Overall Alignment, where the entire model is jointly fine-tuned, the model shows substantial progress in both visual information supplementation and semantic information restoration. For example, compared to the baseline, the score on the visually-focused ChartQA dataset improves by 23.7%, while on the semantically-oriented DocVQA dataset, the improvement reaches 4.2%.

Overall, these experimental results demonstrate that each stage in our proposed training strategy effectively guides the model to learn the corresponding information restoration ability, leading to stable performance gains on the respective test sets.

## 5.6 VISUALIZATION

Figure 3: Visualization of scene text VQA cases.

To further validate the specific role of DRM at different training stages, we incorporate a lightweight pixel decoder during the Visual Alignment stage. This decoder reconstructs image pixels from the enhanced visual tokens output by DRM, supervised by high-quality (HQ) images. Importantly, the gradients from the reconstruction loss are truncated at the output of DRM and only update the decoder's parameters, leaving the Visual Encoder and DRM unaffected. As a result, the decoder itself is not sufficiently trained to produce high-fidelity reconstructions; instead, the reconstructed images mainly serve to highlight the regions that DRM has effectively rectified. As shown in Fig. 2, the outputs reveal how DRM corrects distorted lines or text rows, removes shadows, noise, and highlights, and emphasizes areas containing graphics and text, after Visual Alignment training.

Furthermore, we conducted additional validation of the fully trained model on a scene text question answering task. As illustrated in the figure 3, our model exhibits significantly enhanced robustness under low-quality conditions, such as noise and blur. This further confirms the effectiveness of our proposed dataset and methods.

## 6 CONCLUSION

In this paper, we presented a comprehensive framework to enhance the robustness of MLLMs for low-quality document images. At the core of our method is the DocRobust-Module (DRM), an efficient feature restoration module that recovers lost visual and semantic information with minimal modifications to the overall model parameters, and a two-stage training strategy that incrementally improves the model's restoration ability from both visual and semantic perspectives. To support the training and evaluation of DRM, we constructed a large-scale Visual Question Answering dataset, DocRobust-VQA, which provides diverse low-quality document images along with corresponding QA annotations. This dataset enables scalable training of MLLMs under challenging visual conditions and serves as a valuable resource for robustness research. Extensive experiments demonstrate that our method significantly improves the performance of multimodal large language models on degraded document images, underscoring the potential of targeted restoration techniques to bridge the gap between clear and low-quality inputs. In future work, we plan to further optimize the DRM architecture and training strategies, building on the foundation of the proposed dataset and framework.

ETHICS STATEMENT

This work adheres to the ICLR Code of Ethics. In this study, no human subjects or animal experimentation was involved. All datasets used were sourced in compliance with relevant usage guidelines, ensuring no violation of privacy. We have taken care to avoid any biases or discriminatory outcomes in our research process. No personally identifiable information was used, and no experiments were conducted that could raise privacy or security concerns. We are committed to maintaining transparency and integrity throughout the research process.

REPRODUCIBILITY STATEMENT

We have made every effort to ensure that the results presented in this paper are reproducible. The experimental setup, including training steps, model configurations, and hardware details, is described in detail in the Sec. 5.1. We have also provided a full description of DocRobust-Module in Sec. 4.2, to assist others in reproducing our experiments. Additionally, the dataset used in this work, such as ChartQA Masry et al. (2022), TextVQA Singh et al. (2019), DocVQA Mathew et al. (2021), WildDoc Wang et al. (2025), Inv3dReal Hertlein et al. (2023),etc., are publicly available, and we provided the processing details of generating corresponding low-quality image in 3.2, ensuring consistent and reproducible evaluation results. We believe these measures will enable other researchers to reproduce our work and further advance the field.

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

## A APPENDIX

### A.1 SCALING ABILITY

To further validate the effectiveness of our proposed DRM and two-stage training strategy on larger-scale models, we integrate DRM into InternVL2.5-2B, InternVL2.5-4B, and InternVL2.5-8B and perform the two-stage training consisting of Visual Alignment and Overall Alignment. The test results are presented in Fig 4.

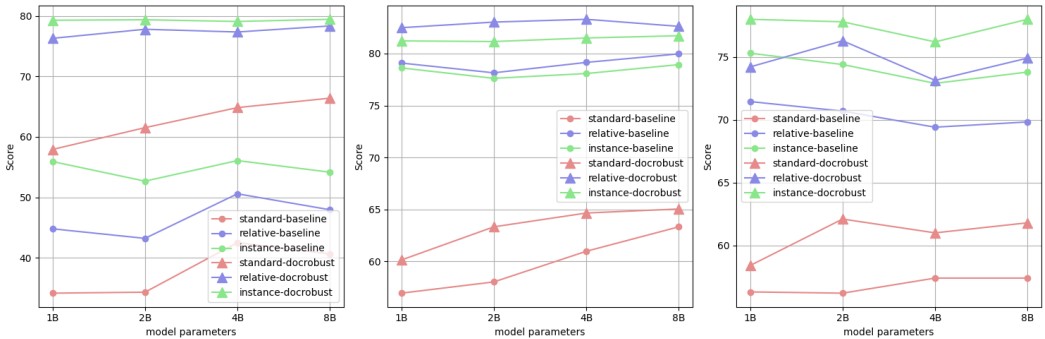

Figure 4: Visualization of the standard, relative, and instance-level relative score from MLLMs of different parameter amounts on ChartQA, TextVQA, and OCRBench. Standard score, relative score, and instance-level relative score reflect the model's absolute accuracy, accuracy on low-quality images, and the per-instance accuracy change after degradation, respectively.

To comprehensively and objectively evaluate the effectiveness of our proposed DRM in enhancing the robustness of MLLMs on low-quality documents, we introduce the following three metrics: Standard Score, Relative Score, and Instance-Level Relative Score, whose detailed definitions are provided below.

### A.1.1 STANDARD SCORE

The Standard Score uses the same evaluation criteria as the original datasets to score the responses of the multimodal model on low-quality document images, reflecting the absolute accuracy. Specifically, for ChartQA, we use relaxed accuracy; for DocVQA and InfographicVQA, we use Average Normalized Levenshtein Similarity (ANLS); for TextVQA, we use the VQA score; and for OCR-Bench, we check whether the ground truth appears in the model's output. This score is formalized as:

$$S_s(X, \text{MLLM}) = F_{data}(\text{MLLM}(X_{cor}), X_{ans})$$

where $F_{data}$ denotes the dataset-specific scoring function, $X_{cor}$ represents the corrupted images, and $X_{ans}$ denotes the answers from the VQA annotations.

### A.1.2 RELATIVE SCORE

It is evident that even on clear document images, the multimodal model may fail to produce the correct output for some images. Counting these inherent errors as being caused by image corruption would not accurately reflect the improvement in robustness. Therefore, we propose the following Relative Score:

$$S_r(X, \text{MLLM}) = \frac{F_{data}(\text{MLLM}(X_{cor}), X_{ans})}{F_{data}(\text{MLLM}(X_{cle}), X_{ans})}$$

where $X_{cle}$ represents the clear images. This metric reflects the ratio of correct answers on corrupted images to those on clear images, providing a more objective measure of the model's enhanced understanding of low-quality images.

### A.1.3 INSTANCE-LEVEL RELATIVE SCORE

In addition to the Relative Score, we propose the Instance-level Relative Score, defined as:

$$S_{ins}(X^j, \text{MLLM}) = \frac{S_s(X^j, \text{MLLM})}{S_r(X^j, \text{MLLM})} - S_s(X^j, \text{MLLM})$$

$$S_{ir}(X, \text{MLLM}) = 1 - \frac{\sum_{j}^{|X|} \mathbb{1}\left[S_{ins}(X^j, \text{MLLM}) > \delta\right]}{|X|} \tag{8}$$

where $\mathbb{1}$ is an indicator function, $S_s(X^j, \text{MLLM})$ and $S_r(X^j, \text{MLLM})$ denote the scores computed on a single image-answer pair $X^j$ in the dataset. Essentially, the term in the summation represents the proportion of images that are correctly understood in their clear state but fail after corruption. A lower ratio indicates stronger robustness, and thus $S_{ir}$ reflects the model's robustness at the instance level.

As shown in 4, it can observed that under different parameter settings, datasets, and scoring configurations, our method consistently outperforms the baseline. This confirms the effectiveness and scalability of our proposed dataset and method. Notably, our DRM underwent Visual Alignment training only within the InternVL2.5-1B visual encoder configuration, without additional training for other model sizes. This demonstrates that the DRM trained with Visual Alignment exhibits a certain degree of scaling ability.

## A.2 COMPUTATIONAL COMPARISON WITH PIXEL-LEVEL RESTORATION

In the Experiment Section (Table 4), we compare the recovery effectiveness of our proposed DRM with representative pixel-level restoration methods. To provide a more holistic evaluation, we further assess the efficiency of each approach in terms of parameter count and computational resource consumption, as reported in Table 5. The results highlight that DRM not only achieves superior robustness enhancement under degraded document conditions, but also offers significant advantages in time and memory efficiency, making it a more practical solution for real-world deployment.

| Method | Params (M) | GFLOPs | Avg. score |
|---|---|---|---|
| InternVL-1B | - | - | 54.28 |
| + DiffBIR (no tune) | 15.8(IR)+1.6k(LDM) | - | 18.60 |
| + DocRes (no tune) | 15.2 | 563.3 | 32.01 |
| + DocRes (finetuned) | 15.2 | 563.3 | 53.32 |
| + DRM (ours) | 56.7 | **29.8** | **61.10** |

Table 5: Average scores and resource cost comparison with pixel-level restoration methods on DocRobust-VQA.

| Method | Tasks | Datasets | | | |
|---|---|---|---|---|---|
| | | ChartQA | TextVQA | DocVQA | InfographicVQA |
| Gemini2.0-flash | clean | 46.76 | 74.30 | 86.70 | 52.92 |
| | corrupted | 41.88 | 65.42 | 78.61 | 45.64 |
| DeepSeek-VL2 | clean | 31.84 | 70.45 | 55.85 | 29.41 |
| | corrupted | 18.48 | 68.43 | 46.04 | 26.78 |
| Claude3.5 | clean | 14.20 | 37.14 | 29.18 | 19.66 |
| | corrupted | 17.36 | 25.99 | 28.59 | 17.24 |
| TextMonkey | clean | 66.92 | 64.06 | 73.10 | 37.76 |
| | corrupted | 36.08 | 47.94 | 61.25 | 31.08 |
| TextHarmony | clean | 66.32 | 67.78 | 64.93 | 40.65 |
| | corrupted | 38.62 | 51.55 | 54.81 | 34.61 |
| mPLUG-DocOwl2.0 | clean | 69.88 | 67.11 | 80.28 | 46.70 |
| | corrupted | 32.08 | 50.95 | 64.18 | 33.79 |

Table 6: More MLLM Results on DocRobust-VQA.

### A.3 MORE MLLM RESULTS ON DOCROBUST-VQA

We further evaluated a broader range of models on DocRobust-VQA, including state-of-the-art proprietary and open-source MLLMs such as DeepSeek-VL2, Claude 3.5, and Gemini 2.0, as well as document-focused models like TextMonkey, TextHarmony, and DocOwl 2.0. As shown in Table 6, all these models exhibit significant performance degradation when tested on the low-quality document images in DocRobust-VQA. This highlights the challenging nature of our synthetic dataset and demonstrates its effectiveness in exposing the robustness limitations of current MLLMs, thereby providing valuable supervision for robustness enhancement.

### A.4 CHOICE OF DRM ARCHITECTURE

In the final design of DRM, we adopt the Transformer architecture as the backbone. Prior to this, we also explored two widely used alternatives—MLP and Mamba. As shown in Table 7, the results on DocRobust-VQA demonstrate that the Transformer-based DRM consistently outperforms its MLP- and Mamba-based counterparts across all evaluation metrics. Based on this empirical evidence, we selected the Transformer as the final architecture for DRM.

## B LLM USAGE

Large Language Models (LLMs) were used to aid in the writing and polishing of the manuscript. Specifically, we used an LLM to assist in refining the language, improving readability, and ensuring clarity in various sections of the paper. The model helped with tasks such as sentence rephrasing, grammar checking, and enhancing the overall flow of the text.

| Method | Tasks | Datasets | | | |
|---|---|---|---|---|---|
| | | ChartQA | TextVQA | DocVQA | InfographicVQA |
| InternVL-1B | clean | 76.24 | 72.01 | 84.75 | 55.75 |
| | corrupted | 34.15 | 56.95 | 72.11 | 37.44 |
| +DRM (MLP) | clean | 75.92 | 71.44 | 84.80 | 56.16 |
| | corrupted | 45.80 | 56.48 | 72.35 | 39.05 |
| +DRM (Mamba) | clean | 75.96 | 71.61 | 84.67 | 56.10 |
| | corrupted | 44.04 | 56.90 | 72.33 | 36.81 |
| +DRM (Transformer) | clean | 75.92 | 72.91 | 84.77 | 56.01 |
| | corrupted | 57.92 | 60.14 | 76.26 | 41.10 |

Table 7: Results of DRM of different architecture on DocRobust-VQA.

It is important to note that the LLM was not involved in the ideation, research methodology, or experimental design. All research concepts, ideas, and analyses were developed and conducted by the authors. The contributions of the LLM were solely focused on improving the linguistic quality of the paper, with no involvement in the scientific content or data analysis.

The authors take full responsibility for the content of the manuscript, including any text generated or polished by the LLM. We have ensured that the LLM-generated text adheres to ethical guidelines and does not contribute to plagiarism or scientific misconduct.

