# OpenReview forum: "DocRobust: Enhancing Robustness of Multi-modal LLMs in Low-Quality Document Image Scenarios"
_ICLR.cc/2026/Conference — ICLR 2026 Conference Withdrawn Submission_

### Official Review · Reviewer_PYGY · 2025-10-25

**Soundness:** 2
**Presentation:** 2
**Contribution:** 2
**Rating:** 4
**Confidence:** 3

**Summary:**

This paper addresses the practical problem of performance degradation in MLLMs when processing low-quality document images. First, the authors introduce the DocRobust-Module (DRM), a lightweight, plug-and-play module designed for feature-level restoration. This module is inserted into the MLLM architecture to recover corrupted visual and semantic information with low parameter overhead. Second, to support the training and evaluation of their approach, the authors construct DocRobust-VQA, a dataset containing over 189K paired high-quality and synthetically degraded document images. Through evaluation, the authors demonstrate that their method improves the performance of MLLMs on the new benchmark as well as on real-world document datasets.

**Strengths:**

1. The DRM is a lightweight, plug-and-play solution that operates efficiently at the feature level. This is a more elegant approach than computationally expensive pixel-level restoration methods.

2. The DRM is trained using a feasible two-stage training strategy (Visual Alignment followed by Semantic/Overall Alignment) that effectively guides the model to restore low-level visual features.

3. The creation of the large-scale DocRobust-VQA dataset provides a much-needed public resource for training and benchmarking model robustness in this domain.

**Weaknesses:**

1. The necessity of the specific two-stage training approach is not fully justified through ablation studies. I observe that in Table 1, the performance of DocRobust-visual and DocRobust-semantic remains bad across most metrics. A crucial baseline is missing: a single-stage, end-to-end fine-tuning of the MLLM with the DRM module (i.e., only the "Overall Alignment" stage).  Consequently, it is unclear whether the Visual Alignment stage and the Semantic Alignment stage are indispensable components for success.

2. The authors do not provide detailed descriptions of the specific procedures for each of the five corruption methods used in generating corrupted images, nor visual examples.

3. Since VQA pairs are derived from clean images, severe corruption may render the ground-truth answers visually unverifiable, thereby introducing noisy signals during training.

4. The following is more of a suggestion than a weakness: The font size of the text in Figure 1 is too small and should be enlarged for better readability.

**Questions:**

1. How would the removal of the Visual Alignment stage and the Semantic Alignment stage impact the performance of DRM?

2. What are the specific implementation details of the five corruption methods used in generating corrupted images?

3. Could excessive corruption occur, resulting in images that cannot be aligned with the Ground Truth?

---

### Official Review · Reviewer_RJRD · 2025-10-26

**Soundness:** 3
**Presentation:** 2
**Contribution:** 2
**Rating:** 4
**Confidence:** 5

**Summary:**

This paper addresses robustness of Multimodal Large Language Models (MLLMs) on low-quality document images. The authors propose DocRobust-Module (DRM), a lightweight feature restoration module inserted between the vision encoder and projector MLP, trained with a two-stage strategy (Visual Alignment followed by Semantic/Overall Alignment). To support training, they construct DocRobust-VQA, a dataset with 189K clear-corrupted image pairs and 417K QA annotations. Experiments on InternVL-2.5 models show consistent improvements on synthetic degradations, real-world images, and even adversarial examples.

**Strengths:**

Robustness to low-quality documents is practically important.
DocRobust-VQA provides a substantial resource (189K pairs, 417K QA)
Consistent improvements across benchmarks.

**Weaknesses:**

Limited diversity in corruption types (only 5 categories).
No analysis of distribution shift between synthetic and real degradations.
Only evaluated on InternVL-2.5 family - generalization to other MLLM architectures unclear.
Missing comparisons with other robustness approaches[1][2][3][4].
[1] UReader [2] DocKylin [3] DocOwl [4] TokenVL [5] Vary

**Questions:**

See the weaknesses.

---

### Official Review · Reviewer_YUNo · 2025-10-30

**Soundness:** 2
**Presentation:** 1
**Contribution:** 2
**Rating:** 2
**Confidence:** 4

**Summary:**

This paper proposes the DocRobust-Module (DRM) to tackle degraded document understanding: DRM integrates with MLLMs for visual/semantic information recovery, uses a two-stage training strategy, constructs the large-scale DocRobust-VQA dataset for fine-tuning, and experiments confirm its consistent performance improvement on low-quality documents.

**Strengths:**

The paper has a good starting point, and the problem it aims to address—the understanding of low-quality document images—is highly meaningful. The proposed method, which focuses on reconstructing high-quality features from low-quality ones, is also logically sound.

**Weaknesses:**

1. The paper focuses on the task of document understanding. However, no document images are presented throughout the entire manuscript—this includes images from both the synthetic dataset and those associated with the DocVQA task.
2. (1) Luminance, (2) Distortion, (3) Blurriness, (4) Noise, and (5) Compression—how is each of these five types of degradation specifically implemented? For document images, Distortion can be introduced to generate camera-captured-like images. However, for scene text images, how should Distortion be incorporated?
3. In Table 4, DocRes is designed for document rectification. What is the significance of applying it to scene text images?

**Questions:**

See the weakness

---

### Official Review · Reviewer_wJxe · 2025-10-31

**Soundness:** 2
**Presentation:** 2
**Contribution:** 2
**Rating:** 4
**Confidence:** 4

**Summary:**

The paper introduces DocRobust-Module (DRM), a lightweight feature restoration module designed to enhance multimodal large language models (MLLMs) for robust document understanding under degradations such as noise, blur, and low resolution. A two-stage training strategy enables DRM to recover lost visual and semantic information efficiently. To support fine-tuning, the authors build DocRobust-VQA, a large-scale dataset with 189K clear-blurry document image pairs and 417K QA annotations. Experiments show that DRM significantly improves MLLM performance on low-quality document images, providing a scalable approach to robust document analysis.

**Strengths:**

- The authors propose to focus on the problem of understanding real-world documents with quality degradation, which is of practical requirement.
- A large-scale dataset is proposed, which consists of 189,771 images paired with 417,502 question-answer pairs, providing sufficient scale and diversity to support the training of multimodal large language models to gain robustness under degraded visual conditions.
- The paper introduces an adapter to improve the quality of visual tokens, which is learned through a three-stage setting.

**Weaknesses:**

- In this paper, only InternVL2.5-1B is used as a baseline for constructing the model, which may constrain the generalization of the proposed method.
- In Table 1, the comparison is somewhat unfair; compared with the InternVL2.5-1B baseline, the better performance may mainly benefit from further data training.
- The technical contribution is relatively not new; the way provided for constructing the model follows a typical way. Besides, how much each stage contributes to the final model is not quite clear.

**Questions:**

Please refer to the weakness part.

---

### Note · Authors · 2025-11-21

I have read and agree with the venue's withdrawal policy on behalf of myself and my co-authors.